# Design Characteristics and Recruitment Rates for Randomized Trials of Peri-Prosthetic Joint Infection Management: A Systematic Review

**DOI:** 10.3390/antibiotics12101486

**Published:** 2023-09-27

**Authors:** Laurens Manning, Bethwyn Allen, Joshua S. Davis

**Affiliations:** 1Medical School, University of Western Australia, Perth, WA 6009, Australia; laurens.manning@uwa.edu.au; 2Infectious Diseases Department, Fiona Stanley Hospital, Murdoch, WA 6150, Australia; 3Library and Information Service for East and South Metropolitan Health Services, South Metropolitan Health Service, Fiona Stanley Hospital, Murdoch, WA 6150, Australia; bethwyn.allen@gmail.com; 4Infection Research Program, Hunter Medical Research Institute, The University of Newcastle, New Lambton Heights, NSW 2305, Australia

**Keywords:** prosthetic joint infections, arthroplasty, randomized controlled trial, trial characteristics, rifampicin, systematic review

## Abstract

Prosthetic joint infections (PJI) present a major management challenge for practicing orthopedic surgeons and infectious disease physicians. There are few high-quality data to inform treatment guidelines. The aim of this systematic review was to report the design characteristics and recruitment rates for randomized controlled trials (RCTs) of PJI management. Trials were considered eligible for inclusion if human participants were randomized to any management intervention for PJI. We searched Medline, PubMed, Embase, Web of Science, Cochrane Database, ANZ Clinical Trials Registry, ClinicalTrials.gov, and the EU Clinical Trials Register until the end of May 2023. The systematic review was registered with PROSPERO (CRD42018112646). We identified 15 published RCTs with a total of 1743 participants with PJI. The median (interquartile range [IQR]) number of successfully recruited participants was 63 (38–140), with 0.28 (0.13–0.96) enrolments per site per month. Only four trials (36.4%) achieved the target recruitment. All RCTs applied different primary endpoints and varying definitions of a ‘good’ outcome. Despite recent improvements, PJI RCTs are characterized by slow recruitment and heterogeneous endpoint assessments, which preclude synthesis in a standard meta-analytic framework. To inform international guidelines, future PJI trials should be run as multi-country trials at high-recruiting sites.

## 1. Introduction

Peri-prosthetic joint infection (PJI) is a complication of joint replacement surgery that occurs in 1–2% of all patients following arthroplasty [1,2], either perioperatively or as a consequence of bacterial seeding of well-fixed implants during bloodstream infections many years after implantation [3,4]. These severe infections result in substantial morbidity to patients in terms of pain, suffering, impaired mobility, prolonged hospitalization, and exposure to broad-spectrum antibiotic therapy. There are substantial societal costs from PJI, such as reduced quality of life, decreased productivity, and direct health care costs [5,6,7]. Reported treatment success varies between 30–90%, and approximately 8% of patients die from PJI within 2 years [6,8]. With an increasing incidence of arthroplasty operations and an increasing prevalence of joint replacements in the wider population [3], particularly in the context of aging populations in most high-income countries, these infections represent a major management challenge for most practicing orthopedic surgeons and infectious disease physicians [9].

PJIs are heterogeneous, with significant differences in presenting features, risk factors, treatment strategies, causative organisms, and outcomes. Clinical phenotypes categorized into early, chronic, and late-acute [4], according to the time from implantation and the duration of symptoms, show particularly striking differences in outcome and recommendations for empiric therapy [10].

Despite the overall burden on the health sector and variable outcomes for individuals, there is a lack of high-quality evidence to inform the management of PJI, with most international guidelines informed by “limited” or “moderate” evidence [10,11,12].

In all but the frailest patients, at least one surgical procedure is essential as an adjunct to antibiotic therapy. The general approach to surgical management of a newly diagnosed PJI is either (i) debridement, antibiotics and implant retention (DAIR), (ii) excision arthroplasty with re-implantation of a new prosthesis at the same operation, followed by a prolonged course of antibiotics (single-stage revision), or (iii) excision arthroplasty, followed by antibiotics and re-implantation of a destination prosthesis later (2-stage revision) [10]. When co-morbidities preclude repeated debridement or revision surgery, PJIs may be managed using excision arthroplasty alone, amputation, or long-term treatment with antibiotics to suppress the infection (with or without preceding debridement).

Antibiotic therapy varies according to the presentation and surgical management strategy. American and Australian guidelines recommend intravenous antibiotics for 2–6 weeks and 6 weeks, [10,13] respectively, followed by oral antibiotics for 3–6 months. The use of rifampicin as a biofilm active agent for staphylococcal PJI was associated with a very large treatment effect in a highly cited but small trial [14], but a follow-up trial that was also underpowered showed no effect [15]. These results sit alongside conflicting observational data, which show no benefit [16] or a small benefit [17] for the use of rifampicin. The efficacy of rifampicin has not yet been demonstrated in an adequately powered randomized controlled trial (RCT).

To our knowledge, there is no consensus on the optimal primary endpoint for RCTs of PJI. In 2018, the international consensus meeting (ICM) for PJI provided a template for outcome reporting, which acknowledged the difficulty of reporting a single dichotomous outcome as a measure of success [18]. Instead, a 4-level ordinal tiered system was proposed, with Tiers 1 and 2 representing control of infection with or without the need for ongoing antibiotics, respectively. Patients in the Tier 3 group represent those for whom a subsequent revision or re-operation was required, which was further subdivided according to any 1 of 6 re-operation types (denoted A–F). Tier 4 represents all-cause mortality, subdivided according to whether the death occurred within 12 months or later. The proposed iCM reporting template also recommended follow-up durations of 1, 5, and 10 years for short-, medium- and long-term follow-up, respectively [18].

Whilst the iCM outcome reporting template may work for benchmarking PJI outcomes within and between institutions, the complexity and range of possible outcomes cannot be applied to a clinical trial framework. Furthermore, this classification approach considers clinical success, the ongoing use of antibiotics, and the need for re-operation and survival but does not capture patient-reported outcome measures (PROMS) such as joint function or quality of life scores. Other approaches to primary endpoints for clinical trials that account for PROMS, as well as traditional measures of ‘treatment success’, have been proposed as the desirability of outcome ranking score (DOOR) but not yet implemented in clinical trials [19].

As part of planning a large-scale clinical trial for PJI, we hypothesized that there was little consistency in primary endpoints across PJI RCTs and that slow recruitment rates were common. The aim of this study was to report the clinical characteristics and recruitment rates of participants with proven or probable PJI who were enrolled in RCTs of management approaches.

## 2. Results

The overall search strategy is provided (Appendix A). The PRISMA flow diagram is shown (Figure 1). From 1396 records identified from the search, 28 were included in the qualitative synthesis. Of these, two trials were registered and withdrawn before recruitment began, two were registered but not yet recruited, and two were registered, unpublished trials with no information available from authors or conference abstracts. One trial was published as an interim analysis [20], with a subsequent publication reporting the entire cohort [21]. One further trial on acute kidney injury from high-dose antibiotic spacers was published but was a subset analysis unrelated to the primary outcome of an ongoing, unpublished trial [22]. The remaining 21 trials included 15 that were published in a peer-reviewed journal and a further six that were unpublished, but data relating to study design and recruitment rates were available from the trial registry or by email contact with the investigators. As of 30 May 2023, a cumulative total of 1743 patients with PJI had been recruited into 15 published randomized trials of clinical management (Table 1, Figure 2). At least another 801 participants had been recruited in ongoing but unpublished clinical trials of PJI (NCT02734134; 363 participants, NCT03435679; 25, NCT02599493; 380, NCT02547129; 4, NCT01667874 17, and NCT04251377, 12).

In total, 1371 and 372 patients have been enrolled in randomized trials assessing antibiotic or surgical management for PJI, respectively. The median trial size was 63 patients, with a range of 14 to 472 (IQR 38–140).

In three (20%) trials, PJI was a subset of all orthopedic implant infections. The remaining 11 (73.3%) trials exclusively recruited patients with PJI. Of these, 6 (54.5%) were for patients with PJI managed with a staged revision. The remaining trials recruited patients with all PJI (*n* = 2), early staphylococcal PJI (*n* = 1), any staphylococcal PJI (*n* = 1), and those PJI managed with debridement and implant retention (*n* = 1).

Five trials assessed surgical management approaches. This included high-pressure lavage, quadriceps snip during the second stage revision of knee PJI, and the impact of closed suction drainage on the elution of antibiotics from impregnated cement. A recent trial is notable as the only published trial comparing broad surgical approaches for PJI, which compared single- versus two-stage revision for hip PJI. The remaining 10 trials were testing different approaches to antibiotic choice (*n* = 4) or antibiotic duration (*n* = 6).

### 2.1. Recruitment Rates

Of the 15 published trials, the median (interquartile range [IQR]) number of successfully recruited participants was 63 (38–140), from a median of 6 (1–17) sites over 37 (26–64) months. The median number of participants recruited from each site per month was 0.28 (0.13–0.96). Amongst those trials exclusively recruiting participants with PJI, only 4 (36.4%) achieved the target recruitment. Three (27.3%) recruited fewer than half of the recruitment target.

### 2.2. Clinical Endpoints

The primary endpoint was ascertained at 12 and 24 months in 5 trials each. The remaining primary endpoints were measured at 5 days, 7 days, 3–6 months, 18 months, and 8–15 years in one trial each.

A dichotomous primary endpoint was chosen for 10 trials, whilst a continuous variable or ordinal primary endpoint was chosen in 4 and 1 trial, respectively. Three of these latter trials had functional outcomes such as range of joint movement, knee society score (KSS), and Western Ontario and McMasters Universities Osteoarthritis Index (WOMAC). One trial applied an ordinal score ranging from 1–7, which ranked outcomes according to survival, clinical cure, and antibiotic-associated adverse events.

Of trials reporting a dichotomous primary endpoint, nine used terms such as cure (3), clinical success (1), or remission (2) to indicate a good outcome or, conversely treatment failure (1) or reinfection (2) to indicate a poor outcome (Table 2). The characteristics that were incorporated into this assessment varied across trials, particularly for the use of C-reactive protein (5 trials), the presence of radiological signs of loosening (2 trials), and the retention of the prosthesis (3 trials). Only one small trial comparing the addition of fusidic acid to rifampicin regimens had ‘antibiotic change’ incorporated into an assessment of clinical success. None of these trials had ongoing antibiotic use as a key characteristic to define a good or poor outcome.

Due to the heterogeneity of primary endpoints, meta-analysis was not performed.

### 2.3. Blinding Outcome Assessment and Randomization

Only one trial was double-blinded. Single blinding of patients or surgeons was reported in one trial each. The remaining trials were reported as ‘open-label’ (9), having ‘no blinding’ of either patients or investigators (2) or not stated (1). Of trials that were reported as ‘open label’, only 3 had outcomes assessed using an adjudication committee blinded to the treatment allocation. One open-label trial used a patient’s self-reported outcome. The process for randomization was reported in 11 trials but not stated in 4.

### 2.4. Sample Size Calculations

Only two trials did not provide any sample size justification. A further trial was explicitly designed as a ‘pilot trial’. A non-inferiority trial design was applied in 5 trials, with stated non-inferiority margins of 5–20%. The remaining trials used a superiority design.

## 3. Discussion

In this systematic literature review, we found that fewer than 2000 patients have ever been enrolled in randomized trials assessing antibiotic or surgical treatment strategies for patients with prosthetic joint infections, and most of these enrolments have been in the past 5 years.

To provide some context for this ‘gap in evidence’, a comparison with non-Hodgkin lymphoma (NHL) is enlightening. The annual incidence of approximately 5000 cases of NHL in Australia is similar [35] to that of PJI [36], with a similarly high survival rate at 5 years. Yet, a review relating to trials of radiotherapy (not chemotherapy) identified nearly 3000 participants from 4 trials alone [37]. Whilst at first, comparisons with radiotherapy for NHL may not appear meaningful, it does highlight that despite a significant burden of disease, clinical trials of PJI are under-represented. This fact helps explain why most clinical guideline recommendations on PJI management are weak and based on low-level evidence.

PJI trials are difficult to perform, and this is reflected in the fact that the median recruitment rate at trial sites was only 3.4 patients per site per year and that only around one-third of trials, including published as well as those that were registered and withdrawn, managed to enroll the target sample size. This is partly due to the fact that, although PJI is common at national and global levels, the annual numbers of patients seen at individual hospitals may be small. To achieve a sufficient sample size to answer important clinical questions accurately, future PJI trials should be run as multi-country trials at high-recruiting sites. Even with a pragmatic platform-trial design with less restrictive inclusion criteria, it is likely that 50–100 sites will be required for a PJI trial larger than 1000 participants over a 2–3 year period.

The primary outcome measure varied substantially between included trials, including in the time frame used and the definitions of cure or treatment failure. Recent publications proposing consensus primary outcomes [18,19] go some way to addressing this problem for future trials, but these outcomes have not yet been road-tested in large, successfully completed trials, and this is needed before they are more widely adopted. The heterogeneity of primary endpoints also precludes quantitative synthesis using standard meta-analytical frameworks.

Ideally, novel primary endpoints could be developed and validated, which build upon a desirability of outcome ranking score (DOOR) that accounts for both patient-reported outcome measures as well as traditional dichotomous measures of treatment success or failure. A clinician-derived DOOR has been proposed for large-scale clinical PJI trials, but should be refined with input from people with lived experience of PJI [19].

This analysis has some limitations. Although our search strategy is reproducible, we may have missed some studies that were not published in English. Second, by prioritizing randomized clinical trials, we did not explore recruitment or outcomes in observational studies.

In a recently published survey assessing expert clinicians’ research priorities in bone and joint infection, four of the five highest-ranked priorities concerned the treatment of prosthetic joint infection [38]. Whilst there has been considerable improvement in trial design and recruitment to PJI trials, in order to address these priorities, larger and better-designed randomized controlled trials are an urgent priority. Now is the time for researchers in this field to learn from history.

## 4. Materials and Methods

The systematic review protocol was prospectively registered with PROSPERO (CRD42018112646; 31 October 2018). Trials were considered eligible for inclusion in the qualitative and/or quantitative synthesis if (human) participants were randomized to any management intervention for PJI. There was no restriction on trial quality, primary outcome, language, or type of intervention. Trials of PJI prevention were excluded. The full search strategy and search terms applied are provided (Appendix A). The following databases were interrogated from the start of each until May 2023: Medline (Ovid, https://www.wolterskluwer.com/en/solutions/ovid/ovid-medline-901, accessed on 29 May 2023), PubMed (for unindexed citations, https://pubmed.ncbi.nlm.nih.gov/, accessed on 29 May 2023), Embase (Ovid, https://www.elsevier.com/solutions/embase-biomedical-research, accessed on 29 May 2023), Web of Science (https://www.webofscience.com/wos/woscc/basic-search, accessed on 29 May 2023), Cochrane Database (including the Trials Register, http://www.cochranelibrary.com/, accessed on 7 June 2023), Australian and New Zealand Clinical Trials Registry (https://www.anzctr.org.au/, accessed on 31 May 2023), ClinicalTrials.gov (https://clinicaltrials.gov/, accessed on 31 May 2023)and the EU Clinical Trials Register (https://www.clinicaltrialsregister.eu/, accessed on 31 May 2023). At the time of data extraction, the WHO International Clinical Trials Registry Platform was not available. Citation records were exported from each search tool and imported into Endnote X9 (Clarivate Analytics) as separate group lists. A combined list of citations was compiled following the manual removal of duplicate records.

Two independent reviewers (LM, JD) screened the titles and abstracts of all identified citations, retrieving full-text copies of those considered relevant for inclusion. Discordant assessments were resolved by consensus. The reviewers were not blinded to study authorship. Studies where PJI were included as either a subset of any implant infection or as a subset of any bone and joint infection were eligible for inclusion. For these studies, data pertaining to PJI were extracted and analyzed. Where trials were registered but unpublished, the investigator responsible for uploading the information to the registry was contacted by email for an update on recruitment.

For each trial, eligibility criteria, intervention, and comparator arms, the primary outcome (including the time point for outcome ascertainment), pre-specified statistical analysis and sample size calculations, target recruitment, actual recruitment, blinding, outcome assessment, randomization procedures, and recruitment rate were extracted where available. The average recruitment rate for each participating site was calculated using the total number of PJIs recruited and the reported start and end dates (in months) divided by the number of participating sites. Data were analyzed using descriptive statistics. No quantitative meta-analysis was planned.

## Figures and Tables

**Figure 1 antibiotics-12-01486-f001:**
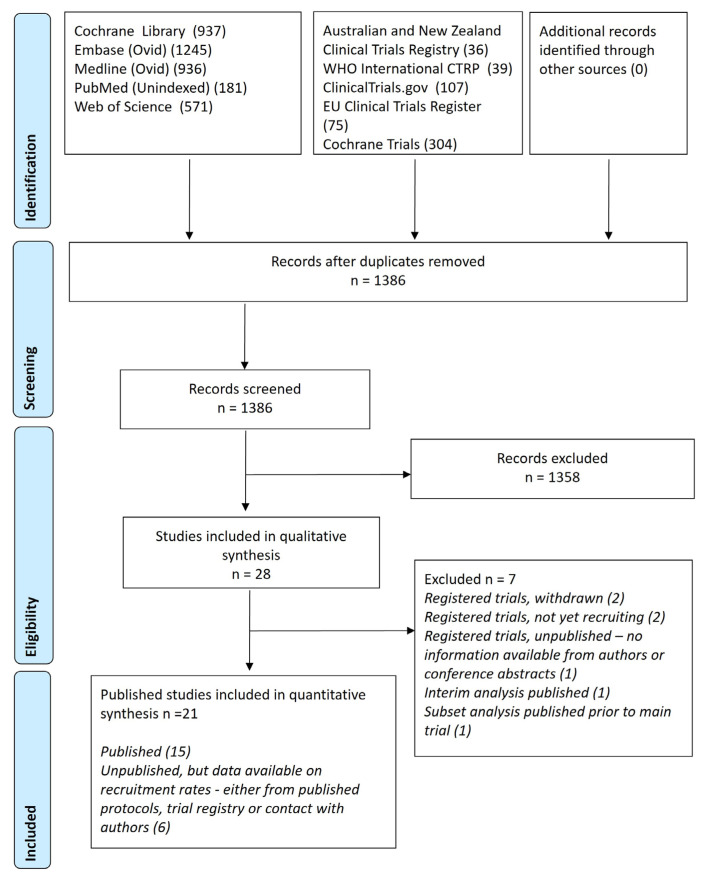
PRISMA figure describing data search and eligible studies for a systematic review of randomized controlled trials for peri-prosthetic joint infection.

**Figure 2 antibiotics-12-01486-f002:**
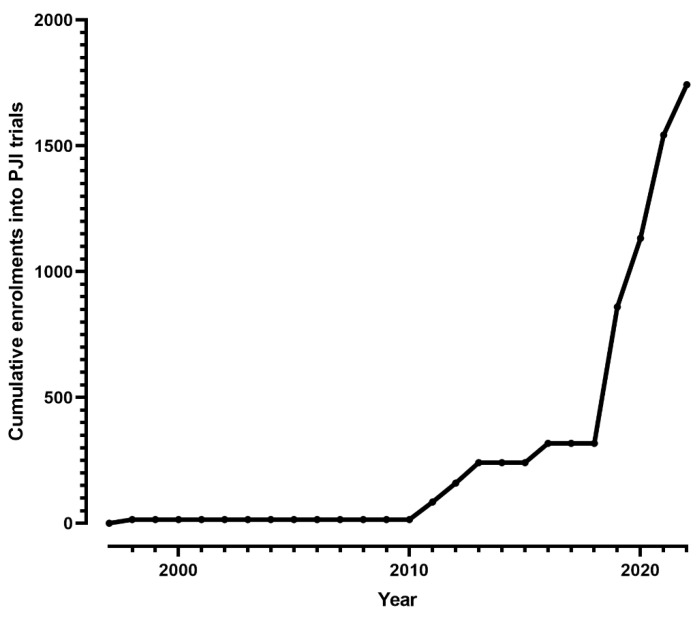
Cumulative total of participants with confirmed peri-prosthetic joint infection (PJI) enrolled in published randomized trials of clinical management.

**Table 1 antibiotics-12-01486-t001:** Design characteristics for randomized trials of periprosthetic joint infection (PJI) management.

Author, Year [Ref]	Trial Registration	Inclusion Criteria	Intervention (s)	Comparator	Primary Outcome	Time Point for Primary Outcome	Sample Size Justification	Target Recruitment (N)	Actual Recruitment (N; Patients with PJI)	Recruitment Period. Sites, N
Zimmerli, 1998 [14]	Not registered	Implant infection (stable implant, short duration)	Rifampicin	Placebo	Cure = no clinical signs, CRP < 5 and no radiology	24 months	Superiority, cure 20% in SOC, 75% in intervention; attrition 20%	30	33 (15 with PJI)	May 1992–1995, 2 centers
Munoz-Mahamud, 2011 [23]	Not registered	Implant infection requiring open debridement	High-pressure lavage	Low-pressure lavage	Remission = no symptoms of infection, prosthesis retained, and CRP < 10	12 months	Not stated	NA	79 (70 with PJI)	January 2008–August 2009, single center
Byren, 2012 [24]	NCT00428844	hip or knee PJI undergoing 2-stage revision	Daptomycin 6 mg/kg Daptomycin 8 mg/k	Vancomycin, teicoplanin, or beta-lactam therapy	Creatinine kinase (CK) >500 IU/mL	Up to 7 days following daptomycin cessation	92% chance of observing at least one occurrence of elevated CK with a true rate of 10%	72 (24 in each arm)	75 (73 evaluable)	June 2007–June 2010, 22 sites
Bruni, 2013 [25]	Not registered	Patients at second stage of 2-stage revision for PJI of knee	Tibial tuberosity osteotomy	Quadriceps Snip	Knee society score (KSS)	Assessments made 8–15 years after procedure	A 20-point difference in the KSS, power of 80%, attrition 15–20%.	90	81 evaluable	1997–2004, single center
Pushkin, 2016 [26]	NCT01756924	Hip and knee PJI (or spacer infection)	Rifampicin and fusidic acid	Standard of care	Success = no evidence of infection, antibiotics not changed	12 weeks for 2-stage, 3–6 months	Not stated	50	14	April 2013–April 2014. 6 centers
Lora-Tamayo, 2016 [27]	ISRCTN35285839	PJI patients managed with DAIR	Short course levofloxacin and rifampicin (8 weeks)	Long course (3 or 6 months)	Cure = retained prosthesis, no clinical signs of infection were resolved, progressive decrease in CRP	12 months	Non-inferiority (NI), 75% expected, NI margin 15%	195	63	April 2009–April 2013, 17 sites
Benkabouche, 2019 [28]	NCT03602209	Implant infections	4 weeks antibiotic	6 weeks antibiotic	Remission of infection at the operative site.	12 months	NI, success rates of 95%, NI margin of 10%.	120	123 (38 with PJI)	1 March 2015 to 10 March 2018, single-center
Li, 2019 [29]	ISRCTN91566927	Bone and joint infections	Short course IV	Long course IV	Definite treatment failure (clinical, micro, histo)	12 months	NI, 5% treatment failure, NI margin 5% (increased to 7.5% during trial)	1050	1054 (472 with PJI))	June 2010–October 2015 recruited, 26 sites
Xu, 2019 [30]	ChiCTR-INR-17014162.	Hip PJI, 2-stage revision with a vanc + mero spacer	Closed suction drain	Non- closed suction drain	Antibacterial activity against MSSA, MRSA, and *Escherichia. coli* during the first five days following spacer implantation.	5 days	NI, >95% antibacterial Activity, difference 20%, power of 80%,	32	32	January–November 2018, Single center
Nahhas, 2020 [31]	NCT01373112	knee PJI undergoing 2-stage revision	Articulating spacer	Static	ROM in degrees	At least 2 years	Superiority, 13 degree difference, power 80% alpha 0.05	68	49 evaluable	July 2011–May 2016, 4 centers
Yang, 2020 [21]	NCT01760863	hip or knee PJI undergoing 2-stage revision	Additional 3 months after 2nd stage	No antibiotics	reinfection as determined by meeting MSIS criteria	“minimum of 2 years”	Superiority; reduction in infection recurrence from 16% to 4%, power 80%, alpha 0.05	200	185	2011–2016, 7 centers
Karlsen, 2020 [15]	NCT00423982	Early or acute Staphylococcal PJI	Rifampicin	Mono	Cure defined as a lack of clinical, biochemical, or radiological signs	24	Superiority; SOC group 70%, intervention 90%, power 80%	200	65 (48 evaluable)	January 2007–June 2013, 8 centers
Bernard, 2021 [32]	NCT01816009	PJI—hip and knee (all)	6 weeks antibiotics	12 weeks antibiotic	The primary endpoint—persistent infection (same organisms) within 2 years	24	NI; expected failure 15%, NI margin 10%	410	410	November 2011–January 2015, 28 centers
Manning, 2022 [33]	ACTRN12617000127303	Acute PJI managed with DAIR	2-weeks IV	6-weeks IV	Seven-level ordinal outcome: Clinical cure will be defined as no clinical or microbiological evidence of infection; original prosthesis still present; and no use of ongoing antibiotic therapy for the index joint	12	Explicitly ‘pilot study’	60	60	June 2017–30 September 2019, 6 centers
Blom, 2022 [34]	ISRCTN10956306	Hip PJI	1-stage	2-stage	Patient-reported WOMAC index	18 months	Difference of 10 points (0.5 SD) 2-sided type 1 error 0.05	148	140	March 2015–December 2018, 15 centers

Abbreviations: CRP—C reactive protein; SOC—standard of care; PJI—prosthetic joint infection; CK—creatinine kinase; DAIR—debridement, antibiotics and implant retention; IV—intravenous; MSSA—methicillin susceptible *Staphylococcus aureus*; MRSA—methicillin resistant *Staphylococcus aureus*; ROM—range of motion; WOMAC—Western Ontario and McMaster Universities Arthritis Index.

**Table 2 antibiotics-12-01486-t002:** Characteristics of dichotomous primary endpoints for treatment outcomes in trials of periprosthetic joint infections.

Study	Timing of Outcome Ascertainment	Terminology	C-Reactive protein (mg/L)	Clinical Features	Radiological Features	Microbiological	Destination Prosthesis	Antibiotic Use/Change
Zimmerli, 1998 [14]	24 months	Cure	CRP < 5	No clinical signs or symptoms of infection	No evidence of loosening or pseudoarthrosis			
Munoz-Mahamud, 2011 [23]	12 months	Remission	CRP < 10	No symptoms of infection/inflammatory signs			Retained	
Pushkin, 2016 [26]	3 months (hip) 6 months (knee)	Clinical success		No clinical signs of infection at re-implantation		Negative cultures obtained at re-implantation		Antibiotics not changed
Lora-Tamayo, 2016 [27]	12 months	Cure	Progressive decline in CRP	No clinical signs of infection, PJI-related death			Retained	
Benkabouche, 2019 [28]	12 months	Remission of infection at the infection site						
Li, 2019 [29]	12 months	Definite treatment failure		Sinus		Isolation of identical organisms, histological		
Yang, 2020 [21]	Minimum of 24 months	Reinfection (MSIS criteria)		Clinical signs		Micro (MSIS)		
Karlsen, 2020 [15]	24 months	Cure	CRP < 10	Lack of clinical signs (e.g., sinus)	No loosening		Retained	
Bernard, 2021 [32]	24 months	Definite or probable infection (persistent or new)	CRP > 10	clinical signs of infection (sinus)		Micro		

## Data Availability

Not applicable.

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
