# Peer review of "Design Characteristics and Recruitment Rates for Randomized Trials of Peri-Prosthetic Joint Infection Management: A Systematic Review"

_antibiotics, 2023, doi:10.3390/antibiotics12101486_

Round 1
Reviewer 1 Report
A nicely written and compact description of the lack of evidence for the management of PJI after arthroplasty and the urgent need to perform large scale international and high quality trials. One of the important contributions is the discussion of definition of endpoints which is very heterogenous in the studies, making comparison of studies, which are already heterogeneous in many factors, even more difficult
I have a few suggestions to further improve the quality of the paper
Interesting data that may be added to the paper (if possible) is the ratio between eligible (screened) patients and actually included patients per study. I thinks this puts the actual inclusion rate per center into more perspective. E.g. the inclusion rate may be low because there were only a few PJI cases, or because only 10% of eligible patients were included).
- Line 136 check for right references
- Line 218: It is stated that only around one third of trials managed to enroll the target sample size. I would say it is sightly better? In Table 1, I counted 7/14 = 50% (one study could not be used here as there was no calculation of target sample size)
- Line 221 Authors state that 50-100 sites will be required for a PJI trial larger than 1000 participants over a 2-3 year period based on the low recruitment rate of 3.4 patients per site per year at trial sites. However, this is dependent on the type of research. If almost all PJI’s fullfill inclusion criteria (e.g. 6 versus 12 weeks for all PJI) inclusion will be much easier compared to trials in which only a subgroup of patients with PJI are eligible for inclusion (e.g. only staphylococaal PJI after DAIR). Further, the number of performed arthroplasties per center contribute significantly. Therefore, than number of 3.4pts per year does not say that much and could be more nuanced.
- Line 236. The meta-analysis has not been mentioned yet in sections above.
- Line 236: I would not consider it a weakness (but instead consider it the right choice) that a formal meta-analysis has not been performed as the studies are quite heterogeneous with respect to inclusion criteria and objectives which ha s high impact on inclusion rate. A meta-analysis of recruitment rates would therefore in my opinion not result in a clinical relevant conclusion to this paper
Reviewer 2 Report
Those of us in Infectious Diseases and Orthopedics who care for patients with PJI spend a large amount of time parsing the vast retrospective and sparse RCT literature to answer important and common clinical questions. Despite the frequency and importance of PJI, it remains relatively understudied compared to other disciplines, in part because there is no marketing incentive for optimizing PJI management. The authors seek to understand the scope of and limitations within PJI research by identifying and summarizing the available RCTs that evaluated treatment for PJI. This important study identifies three important findings: First it highlights the relative research gap in PJI versus other illnesses. Further this study demonstrates that recruitment of patients into PJI studies has been insufficient and contributes to an overall lack of power in the published literature. Finally this study identified the heterogeneity in RCT study outcomes which challenges one’s ability to compare results between studies.
The authors‘ review was comprehensive, and did not miss any studies that I was separately aware of. The search strategy and methodology was sound, as was the manuscript text and tables/figures. They importantly tabulated the differences in design, calling forward the importance of more standardization going forward. I especially appreciated Figure 2, which demonstrates the more recent growth in the number of included PJI study participants, which does provide some hope for the future.
I have no major concerns with this paper and have only one suggestion which would be to be more concrete in the call to action. What could the relevant societies (e.g. EBJIS and MSIS) do to improve these deficiencies? If the ICM outcomes criteria are insufficient for RCTs, should a separate group be convened to develop a standardized outcome for this type of study, which would also include PROMs? What about funding for the development of infrastructure and international collaboratives that could support multiple RCT studies over time?
Minor
1. Intro line 34-35: the authors indicate that early infection is due to contamination during the index operation. I don’t believe this is fully known – it is possible that the contamination may occur in the early postoperative period due to early wound healing complications before the deep joint healing is secure. Consider rewording.
2. The PRISMA 2009 flow diagram (supplement) is incomplete with (n= )
